# MODEL FUSION OF HETEROGENEOUS NEURAL NETWORKS VIA CROSS-LAYER ALIGNMENT

## ABSTRACT

Layer-wise model fusion via optimal transport, named OTFusion, applies soft neuron association for unifying different pre-trained networks to save computational resources. While enjoying its success, OTFusion requires the input networks to have the same number of layers. To address this issue, we propose a novel model fusion framework, named *CLAFusion*, to fuse neural networks with a different number of layers, which we refer to as heterogeneous neural networks, via cross-layer alignment. The cross-layer alignment problem, which is an unbalanced assignment problem, can be solved efficiently using dynamic programming. Based on the cross-layer alignment, our framework balances the number of layers of neural networks before applying layer-wise model fusion. Our synthetic experiments indicate that the fused network from CLAFusion achieves a more favorable performance compared to the individual networks trained on heterogeneous data without the need for any retraining. With an extra finetuning process, it improves the accuracy of residual networks on the CIFAR10 dataset. Finally, we explore its application for model compression and knowledge distillation when applying to the teacher-student setting.

## 1 INTRODUCTION

The ubiquitous presence of deep neural networks raises the question, "Can we combine the knowledge of two and more deep neural networks to improve performance?" As one of the earliest successful methods, ensemble learning aggregates the output over a collection of trained models to improve the generalization ability (Hansen & Salamon, 1990; Zhou, 2012). However, it is expensive in terms of computational resources because it requires storing and running many trained neural networks during inference. To overcome the previous hardness, we need to find a single neural network that can inherit the knowledge from multiple pre-trained neural networks and have a small size at the same time. To our best knowledge, there are two popular approaches for this purpose: knowledge distillation and model fusion.

The first approach, *knowledge distillation*, is a machine learning technique that transfers knowledge from large networks (teachers) into a smaller one (student) (Hinton et al., 2015). The smaller network is trained by the task-specific loss and additional distillation losses that encourage the student networks to mimic the prediction of the teachers. The distillation process, however, is computationally expensive because it trains the student network from scratch. In addition, at each training epoch knowledge distillation runs forward inference for all networks and demands sharing the large training data. In some situations sharing data is prohibited due to critical issues such as data security or network bandwidth constraints.

The second line of works is called *model fusion* which is the problem of merging a collection of pre-trained networks into a unified network (Wang et al., 2020; Singh & Jaggi, 2020). The simplest technique in model fusion is vanilla averaging (Utans, 1996; Smith & Gashler, 2017), which computes the weighted average of pre-trained network parameters without the need for retraining. Their methods fuse neural networks with the same architecture without considering the permutation invariance nature of neural networks. To mitigate this problem, the idea of solving a neuron alignment problem before applying weight averaging is discussed in Ashmore & Gashler (2015); Li et al. (2015). Recently, there are two concurrent works that benefit from the idea of neuron alignment. FedMA (Wang et al., 2020) formulated and solved the assignment problem to find a permutation

matrix that undoes the permutation of weight matrices. While OTFusion (Singh & Jaggi, 2020) viewed the problem through the lens of optimal transport (Monge, 1781; Kantorovich, 2006) and utilized the transport map to align the weight matrices. Though both methods can fuse multiple neural networks, their applications are limited to networks with the same number of layers.

**Contributions.** In this paper, we propose a model fusion framework for fusing heterogeneous neural networks, namely, *unequal width and unequal depth* neural networks, which is named as *Cross-Layer Alignment Fusion* (CLAFusion). CLAFusion consists of three parts: cross-layer alignment, layer balancing method, and layer-wise model fusion method. We first formulate the cross-layer alignment as an assignment problem using layer representation and layer similarity, then propose a dynamic programming-based algorithm to solve it. Next, we present two natural and fast methods to balance the number of layers between two networks. Finally, our experiments demonstrate the efficiency of CLAFusion on three different setups. In skill transfer, our framework successfully fuses two heterogeneous neural networks trained on heterogeneous data and improves the performance over the individual networks without retraining. In addition, the fused model from CLAFusion serves as an efficient initialization when training residual networks. Furthermore, CLAFusion shows potential applications for model compression and knowledge distillation in the teacher-student setting.

## 2 BACKGROUND

In this section, we first recall the layer representation and layer similarity that have been widely used in applications. Then, we review available model fusion methods and their limitations. Finally, we discuss OTFusion and the challenge of applying this framework to heterogeneous neural networks.

### 2.1 LAYER REPRESENTATION AND LAYER SIMILARITY

A common layer representation is the matrix of activations (Kornblith et al., 2019). Activation matrix, also known as activation map, can be used as a type of knowledge to guide the training in knowledge distillation (Gou et al., 2021). Another choice is the weight matrix of the trained neural networks, which has been shown to perform well on intra-network comparison tasks Neill et al. (2020).

The neural network exhibits the same output when its neurons in each layer are permuted, this is so-called the permutation invariance nature of neural network parameters. Due to the permutation invariance property, a meaningful layer similarity index should be invariant to orthogonal transformations (Kornblith et al., 2019). Their proposed Centered Kernel Alignment (CKA) effectively identifies the relationship between layers of different architectures.

### 2.2 MODEL FUSION

Despite showing good empirical results, vanilla averaging (Utans, 1996; Smith & Gashler, 2017) only works in the case when the weights of individual networks are relatively close in the weight space. As an effective remedy, several works on model fusion considered performing permutation of neurons in hidden layers before applying vanilla averaging. FBA-Wagging (Ashmore & Gashler, 2015), Elastic Weight Consolidation (Leontev et al., 2019), and FedMA (Wang et al., 2020) formulate the neuron association problem as an assignment problem and align the neurons. Nevertheless, those variants are not generalized to heterogeneous neural networks.

There is limited literature devoted to the discussion of fusing heterogeneous neural networks. NeuralMerger (Chou et al., 2018) is one of the few attempts to deal with this setting. However, there are two key differences in NeuralMerger from our CLAFusion. First, their cross-layer alignment is hand-crafted and dedicated to specific neural network architectures. Secondly, they decompose weights into lower dimensions and use vector quantization to merge the weights of two networks. On the other hand, we introduce a systematic way to solve the cross-layer alignment problem, and combining weights is done using the layer-wise model fusion method.

### 2.3 MODEL FUSION VIA OPTIMAL TRANSPORT

Recently, Singh & Jaggi (2020) proposed OTFusion, which solves the neuron association problem based on free-support barycenters (Cuturi & Doucet, 2014). OTFusion leads to a noticeable improve-

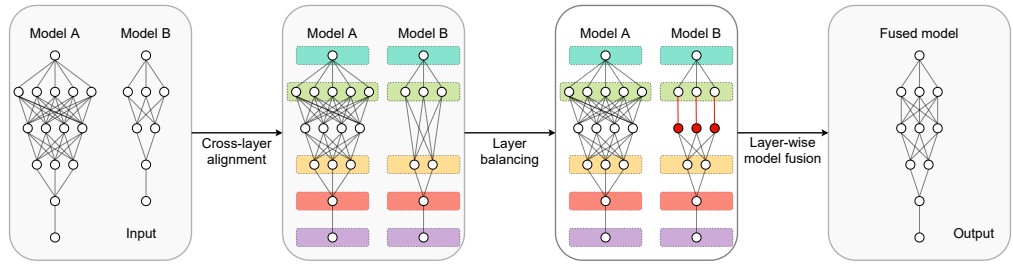

Figure 1: **CLAFusion strategy:** In the first step, the cross-layer alignment problem is solved for two pre-trained models. Two corresponding layers are surrounded by two rounded rectangles of the same color. Based on the optimal mappings obtained in the previous step, CLAFusion balances the number of layers (adding layers in this figure). The red circles and lines indicate the newly added neurons and weights (Zero weights are omitted). Finally, CLAFusion applies a layer-wise model fusion method to produce the final output.

ment over vanilla averaging when all individual networks have the same architecture. One advantage of OTFusion over other vanilla averaging-based model fusion methods is that it can fuse networks with different widths (i.e., number of neurons) thanks to the nature of optimal transport.

**The challenge of applying OTFusion to heterogeneous neural networks.** OTFusion, however, is still a layer-wise approach that cannot be directly employed in heterogeneous neural networks setting. We need to find two corresponding layers, matching their neurons before averaging weight matrices. There are two challenges to this scheme. Firstly, the one-to-one mapping between layers is not available in advance. A naive one-to-one mapping of layers with the same index as the layer-wise approach may cause a mismatch when the numbers of layers are different. For example, we analyze two VGG configurations (Simonyan & Zisserman, 2014, Table 1) with different depths: VGG11 and VGG13. The second convolution layer of VGG11 has $128$ channels and an input image of size $112 \times 112$. While that of VGG13 has $64$ channels and an input image of size $224 \times 224$. Secondly, assume that we have successfully found a one-to-one mapping, there still necessitates a special treatment for remaining layers that has no counterparts. Simply removing those layers has an adverse effect on the deeper network, this may cumulatively degrade the performance of the fused model.

## 3 CROSS-LAYER ALIGNMENT MODEL FUSION

To overcome the challenges of OTFusion, in this section, we propose Cross-layer Alignment Model Fusion (in short, *CLAFusion*) framework for fusing heterogeneous neural networks.

**Notation.** We define $[\![n]\!]$ as a set of integers $\{1, \ldots, n\}$ and $\mathbb{I}_n$ as an identity matrix of size $n \times n$. We use A, B to denote the individual models and $\mathcal{F}$ to denote the fused model. $l$ indicates the layer index. Layer $l$ of model A has $p_A^{(l)}$ neurons and an activation function $f_A^{(l)}$. The pre-activation and activation vector at layer $l$ of model A are denoted as $\boldsymbol{z}_A^{(l)}, \boldsymbol{x}_A^{(l)} \in \mathbb{R}^{p_A^{(l)}}$, respectively. The weight matrix between layers $l$ and $l-1$ of model A is $\boldsymbol{W}_A^{(l,l-1)}$. The following equations hold between two consecutive layers of model A (we omit the bias terms here).

$$\boldsymbol{x}^{(l)} = f_A^{(l)}(\boldsymbol{z}_A^{(l)}) = f_A^{(l)}(\boldsymbol{W}_A^{(l,l-1)} \boldsymbol{x}^{(l-1)}). \tag{1}$$

We stack the pre-activation vector over $t$ samples to form a $t-$row pre-activation matrix. Let $\boldsymbol{Z}_A^{(l)} \in \mathbb{R}^{t \times p_A^{(l)}}$ denote a matrix of pre-activations at layer $l$ of model A for $t$ samples, and $\boldsymbol{Z}_B^{(l)} \in \mathbb{R}^{t \times p_B^{(l)}}$ denote a matrix of pre-activations at layer $l$ of model B for the same $t$ samples. The activation matrices $\boldsymbol{X}_A^{(l)}, \boldsymbol{X}_B(l) \in \mathbb{R}^{t \times p_B^{(l)}}$ are obtained by applying the corresponding activation functions element-wise to the pre-activation matrices.

**Problem setting.** Hereafter, we consider fusing two feed-forward networks A and B of the *same architecture family*. Two networks have the same input and output dimension but a different number of hidden layers. In each network, the hidden layer has an activation function of ReLU, which is the general setting in modern architectures. Additional work is required to handle bias terms and the batch normalization layer properly so we leave them for future work. Let $m$ and $n$ be the number of hidden layers of models A and B, respectively. Taking the input and output layers into account, the

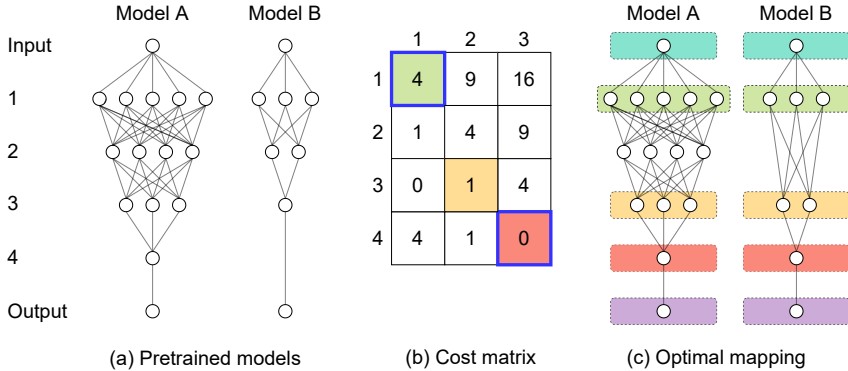

Figure 2: **Cross-layer alignment example:** Two pre-trained neural networks are given in (a). Model A has 4 hidden layers of size 5, 4, 3, 1 while model B has 3 hidden layers of size 3, 2, 1. (b) shows the cost matrix (squared Euclidean distance) between layer representations (number of neurons) for hidden layers of two networks. Three color cells represent the solution of the CLA problem. Note that the upper-left and lower-right cells are automatically chosen by the constraints on the first and last hidden layers. (c) visualizes the optimal mapping. Two rounded rectangles of the same colors represent two corresponding layers in the optimal CLA.

numbers of layers are $m + 2$ and $n + 2$, respectively. Without loss of generality, assume that $m \geq n$. The layer index of model A and B are $\{0, 1, \ldots, m + 1\}$ and $\{0, 1, \ldots, n + 1\}$, respectively.

**General strategy.** Our framework has three components, which are illustrated in Figure 1. The first part is a cross-layer alignment which is a one-to-one mapping from hidden layers of model B to hidden layers of model A. The second part is a layer balancing method that equalizes the number of layers of two models based on the cross-layer alignment. The last part is a layer-wise model fusion method that is applied to same-sized models. The third part of our framework adopts any model fusion methods that can fuse two neural networks with the same number of layers. Next, we give more details about the first and second parts as well as provide some concrete methods.

### 3.1 CROSS-LAYER ALIGNMENT

Cross-layer alignment (CLA) is a nontrivial problem that also arises in the context of the feature-based knowledge distillation approach (Zagoruyko & Komodakis, 2016; Tung & Mori, 2019; Chen et al., 2020). CLA requires finding a one-to-one function that matches similar layers between two models. Out of a few model fusion approaches considering CLA, NeuralMerger is based on the hand-crafted layer association, which causes a lack of generalization. To address this matter, we introduce an efficient method for solving the CLA problem.

Because the input and output layers of the two models are identical in both dimension and functionality, we only align their hidden layers. In addition, the first and last hidden layers usually play a critical role in the model performance (Zhang et al., 2019). Therefore, we directly match the first and last hidden layers of the two models. The importance of this constraint in CLA is studied in Appendix C. Furthermore, two networks are feed-forward, so the mapping is necessary to keep the sequential order of layers. Taking all into consideration, we formulate the CLA problem as follows.

**Definition 1.** *Assume that we have two layer representations for hidden layers of two models as,* $\boldsymbol{L}_A = \{L_A^{(1)}, \ldots, L_A^{(m)}\}$ *and* $\boldsymbol{L}_B = \{L_B^{(1)}, \ldots, L_B^{(n)}\}$. *Let* $\boldsymbol{C}_{i,j}$ *denote the cost matrix between them, i.e.,* $\boldsymbol{C}_{i,j} = d(L_A^{(i)}, L_B^{(j)})$ *where* $d$ *is a layer dissimilarity. The optimal CLA is a strictly increasing mapping* $a : [\![n]\!] \mapsto [\![m]\!]$ *satisfying* $a(1) = 1, a(n) = m$ *and minimizing* $\sum_{i=1}^{n} \boldsymbol{C}_{a(i),i}$.

**Discussion.** The above problem is a special case of the (linear) unbalanced assignment problem (Ramshaw & Tarjan, 2012). The condition of strictly increasing ensures the sequential order of layers when matching. If we have two increasing layer representations in 1-D, the problem can be interpreted as the Partial Optimal Transport in 1-D problem with uniform weights. It can be solved efficiently in $O(mn)$ using the proposed method in Bonneel & Coeurjolly (2019). However, the increasing constraint on layer representations is quite strict and layer representations might not be in 1-D in general. On the other hand, the unbalanced assignment problem can be solved using the generalization of the Hungarian algorithm (Kuhn, 1955). The time complexity of the Hungarian algorithm in our CLA problem is $O(mn^2)$.

---

**Algorithm 1** Cross-layer alignment algorithm

---

    **Input:** $\boldsymbol{C} = [\boldsymbol{C}_{i,j}]_{i,j}$
    $S(i,i) \leftarrow \sum_{l=1}^{i} \boldsymbol{C}_{l,l}, i \in [\![n]\!]$
    $S(0,j) \leftarrow 0, j \in [\![m]\!]$
    $S(i,0) \leftarrow 0, i \in [\![n]\!]$
    **for** $i = 1$ **to** $n$ **do**
        **for** $j = i + 1$ **to** $m$ **do**
            $S(i,j) \leftarrow \min\{S(i,j-1), S(i-1,j-1) + \boldsymbol{C}_{j,i}\}$
        **end for**
    **end for**
    $a(1) \leftarrow 1; a(n) \leftarrow m; i \leftarrow n - 1; j \leftarrow m - 1$
    **while** $i \geq 2$ **do**
        **while** $j \geq i + 1$ and $S(i,j) = S(i,j-1)$ **do**
            $j \leftarrow j - 1$
        **end while**
        $a(i) = j; i \leftarrow i - 1; j \leftarrow j - 1$
    **end while**
    **Output:** $a$

---

**Cross-layer alignment algorithm.** We propose an efficient algorithm based on dynamic programming to solve the CLA problem. Algorithm 1 details the pseudo-code for our algorithm. Given the cost matrix, the time complexity of Algorithm 1 is only $O(mn)$. Note that the optimal alignment from Definition 1 is not necessarily unique so we choose one which is obtained by backtracking.

**Layer representation.** An important process of computing the cost matrix is to design an appropriate representation for the hidden layer. We list three possible layer representations for hidden layers of two models. The first representation is the number of neurons in each layer. This encourages two layers that have a similar number of neurons to match with each other. The second choice is the activation matrix which is widely used for cross-layer comparison. We can also use the pre-activation matrix instead of the activation matrix. The third alternative is the weight matrix of the pre-trained model. Next, we discuss the choice of the cost function (layer dissimilarity). For the first representation, either the Euclidean distance or its squared version is sufficient. To measure the discrepancy between two activation matrices we choose the dissimilarity index based on CKA while another option is the Wasserstein distance. Finally, the cosine distance, Kullback–Leibler divergence, and Wasserstein distance can be used in the weight matrix representation (Neill et al., 2020).

**CLA for Convolutional neural network (CNN).** Following NeuralMerger, we do not match fully connected layers and convolution layers. It is natural that a convolution layer has different functionality from a fully connected layer. In addition, there is a lack of an appropriate way to combine the weights of a fully connected layer and a convolution layer. For the rest of this paper, we consider CNN architecture which consists of consecutive convolution layers followed by fully connected layers such as VGG (Simonyan & Zisserman, 2014) and RESNET (He et al., 2016). The key idea is to solve the CLA for same-type layers separately then combine two mappings. Note that the combination of two strictly increasing mappings is also a strictly increasing mapping, thus it satisfies the strictly increasing requirement. We further break the convolution layers part into smaller groups and find the mapping for each pair of groups. In particular, we divide into 5 groups separated by the max-pooling layers in VGG. In RESNET, we group consecutive blocks with the same number of channels, also known as stage, and in each stage, align blocks instead of layers. For the block representation, we choose the (pre-)activation matrix of the second convolution layer in each block.

## 3.2 LAYER BALANCING METHODS

As mentioned above, some layers in model A may have no counterparts in model B in the optimal mapping. Naturally, we have two opposite directions: reduce layers in model A or add layers into model B. Assume that we have already balanced the number of layers up to layer $l$ of model B. At layer $l + 1$ of model B, there are two possibilities. If $a(l + 1) - a(l) = 1$, the layer-wise fusion method is ready to apply up to layer $l + 1$. If $a(l+1) - a(l) > 1$, we can either merge layers between

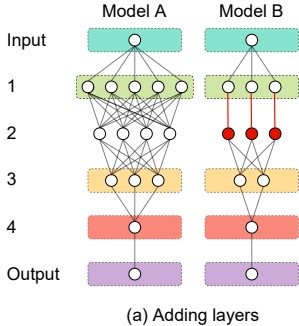 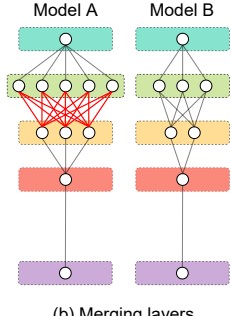

|  | (a) Adding layers | (b) Merging layers |

Figure 3: **Layer balancing examples:** In (a), a new layer is added between hidden layers 1 and 2 of model B. Newly added neurons and weights are in red (We omit zero weights). The new weight matrix between layers 1 and 2 of model B is an identity matrix whose size equals the number of neurons at layer 1. While the old weight matrix between layers 1 and 2 in model B becomes the new weight matrix between layers 2 and 3. In (b), hidden layer 2 of model A is removed. The red lines indicate the new weight matrix which is calculated based on two old weight matrices and the activation matrix of the removed layer.

$a(l)$ and $a(l+1)$ of model A or add layers between $l$ and $l+1$ of model B. Next, we discuss two approaches in the case $a(l+1) - a(l) = 2$. When $a(l+1) - a(l) > 2$, we can repeat the same approach $a(l+1) - a(l) - 1$ times.

**Add layers into model B.** We add a layer $l'$ between layer $l$ and $l+1$ of model B. The new weight matrices are defined as $\boldsymbol{W}_B^{(l',l)} \leftarrow \mathbb{I}_{p_B^{(l)}}$ and $\boldsymbol{W}_B^{(l+1,l')} \leftarrow \boldsymbol{W}_B^{(l+1,l)} \in \mathbb{R}^{p_B^{(l+1)} \times p_B^{(l)}}$. The new activation function is defined as $f_B^{(l')} \leftarrow f_A^{(a(l)+1)}$, which is ReLU. Because $x_B^{(l)} \succeq 0$ for all $l \in [\![n]\!]$, from Equation 1 we have

$$x_B^{(l')} = f_B^{(l')}(\boldsymbol{W}_B^{(l',l)} x_B^{(l)}) = ReLU(x_B^{(l)}) = x_B^{(l)}.$$

Therefore, the information of model B remains unchanged after adding layer $l'$. Note that the new layer is just an identity mapping, which is a trick that has been used in RESNET and NET2NET (Chen et al., 2015). Also discussed in NET2NET, the adding layers method is a function-preserving transformation that allows us to generate a valuable initialization for training a larger network. We later examine this hypothesis in our experiments.

**Merge layers in model A.** We merge layer $a(l)+1$ into layer $a(l)$ of model A by directly connecting layer $a(l)$ to layer $a(l+1)$. Because $f_A^{(a(l)+1)}$ is ReLU, the new weight matrix can be written as

$$\boldsymbol{W}_A^{(a(l+1),a(l))} = \boldsymbol{W}_A^{(a(l+1),a(l)+1)} \boldsymbol{D}_A^{(a(l)+1)} \boldsymbol{W}_A^{(a(l)+1,a(l))},$$

where $\boldsymbol{D}_A^{(a(l)+1)} \in \mathbb{R}^{p_A^{(a(l)+1)} \times p_A^{(a(l)+1)}}$ is an *input-dependent* diagonal matrix with 0s and 1s on its diagonal. The $i^{th}$ entry in the diagonal has a value of 1 if the $i^{th}$ entry of $\boldsymbol{z}_A^{(a(l)+1)}$ is positive and 0 otherwise. Because the actual sign of neuron $i$ at layer $a(l)+1$ varies by the input, we provide a simple estimation. Given that the neuron $i \in [\![p_A^{(a(l)+1)}]\!]$ has a pre-activation vector over t samples as $\boldsymbol{z}_{.,i} \in \mathbb{R}^t$, which is the $i^{th}$ column of the pre-activation matrix $\boldsymbol{Z}_A^{(a(l)+1)}$. We estimate the sign of neuron $i$ using either sign of sum (sgn $\sum_{j=1}^t \boldsymbol{z}_{j,i}$) or sign of majority (i.e., 1 if at least $t/2$ samples have positive activation values, 0 otherwise).

**Pros and cons of balance methods.** An advantage of merging layers is that the fused model has fewer layers. However, merging layers degrades the accuracy of model A and it is slower than adding layers method that does not involve any complex calculations. On the flip side, adding layers does not affect the accuracy of model B but results in a deeper fused model. Surprisingly, merging layers shows comparatively better performance than adding layers in our experiments on MLP. Performance comparison between the two methods will be provided in Appendix C and D.1. Note that we can use more sophisticated model compression methods instead of the merging layers method. Similarly, it is possible to replace the adding layers method with a more advanced network expanding technique (Wei et al., 2016). Here, we just introduce two natural and fast ways to demonstrate our framework.

**Balancing the number of layers for CNN.** The same merging method does not work in the case of CNN. On the other hand, the adding layers method can be easily applied for convolution layers.

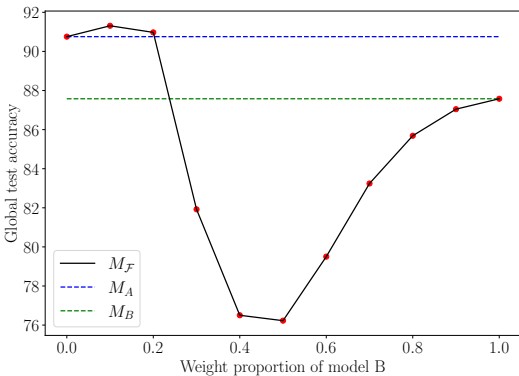

Figure 4: The performance (global test accuracy on the whole MNIST dataset) of the fused model when fusing the specialist model A and the generalist model B in various weight proportions of model B. All results are averaged across 5 seeds. Detailed results are given in Table 8.

For VGG, we can set all filters of a new convolution layer to identity kernels. For RESNET, we add a new block in which all filters of two convolution layers become zero kernels while the short-cut connection remains as the identity mapping. A block without a short-cut connection can be simply replaced by two identity convolution layers as in VGG.

## 4 EXPERIMENTS

**Outline.** Our method is first applied for two MLPs trained on the synthetic MNIST dataset (LeCun et al., 1998). We then fuse a RESNET34 and a RESNET18 trained on CIFAR10 dataset (Krizhevsky et al., 2009). The last one is a teacher-student setting in which VGG architecture is utilized. The detailed settings including training hyperparameters, used assets, and computational resources are specified in Appendix B. We further conduct ablation studies to validate the effectiveness of cross-layer alignment and compare two layer balancing methods in Appendix C. The experimental results for different seeds can be found in Appendix D.

### 4.1 SKILL TRANSFER

**Settings.** In skill transfer, we aim to obtain a single model that can inherit both overall and specialized skills from the two individual models. We adopt the same heterogeneous data-split technique as in (Singh & Jaggi, 2020, Section 5.1) for two MLPs A and B. The MNIST dataset is separated into two datasets. One dataset has all images of label 4 and 10% images of other labels while the other contains the rest. We train a MLP with 3 hidden layers of size 400, 200, 100 for model A and a MLP with 4 hidden layers of size 400, 200, 100, 50 for model B. The pre-activation matrix is obtained by running inference for 400 samples. We use the number of neurons as layer representation and adding layers as the balancing method. The weight proportion of model B, which is referred to as $w_B$, is swept over 11 different values: $w_B = \{0.0, 0.1, \ldots, 1.0\}$.

**Results.** Figure 4 shows that when the weight proportion of model B is small (0.1 or 0.2), the fused model improves over both individual models. The reason is that the specialist model A has higher accuracy than the generalist model B. When distributing the weight proportion fairly (0.5), the fused model performs worst due to the effect of heterogeneous training data. The fused model has the best accuracy of 91.31%, which is higher than those of models A (90.76%) and B (87.58%). This claims that the knowledge has been transferred successfully to the fused model without retraining.

### 4.2 GENERATE AN EFFICIENT INITIALIZATION

**Settings.** Following the experiments conducted in (Singh & Jaggi, 2020, Section 5.3), CLAFusion can be used to generate an initialization when training the larger neural network. We apply CLAFusion to pre-trained RESNET34 and RESNET18 on CIFAR10 dataset. Layer representation is the pre-activation matrix of 200 samples while the cost function is calculated by subtracting the linear CKA from 1. After fusing, we also perform finetuning to improve the accuracy as observed in previous

| $M_A$ | $M_B$ | Ensemble | $M_{\mathcal{F}}$ | Finetune | | | | |
|-------|-------|----------|-------------------|----------|---|---|---|---|
| | | Learning | | $M_A$ | $M_B$ | $M_{\mathcal{F}}$ | $M_B$ depth-aligned | Random RESNET34 |
| 93.31 | 92.92 | 93.81 | 65.58 | 93.52 | 93.29 | **93.60** | 93.22 | 92.00 |
| ±0.13 | ±0.20 | ±0.21 | ±2.61 | ±0.09 | ±0.14 | ±0.14 | ±0.13 | ±0.17 |

Table 1: The results of fusing and finetuning residual networks on CIFAR10 dataset. The results shown are mean ±std. deviation over 5 seeds. The detailed results are reported in Table 11.

| Method | HMT | HMT + Naive avg | HMT + OTFusion | HMT + CLA + OTFusion |
|--------|-----|-----------------|----------------|----------------------|
| Transfer | 10.79 ±1.01 | 37.49 ±4.59 | 18.32 ±1.76 | **55.22** ±3.30 |
| Finetune | 92.93 ±0.34 | 91.34 ±4.00 | 93.66 ±0.14 | **93.70** ±0.05 |

Table 2: The results of heterogeneous model transfer from RESNET18 to RESNET34. Experiments are run 5 times and details can be found in Table 12.

model fusion works (Chou et al., 2018; Singh & Jaggi, 2020). The finetuning hyperparameters, which are adopted from (Singh & Jaggi, 2020, Section S3.1.3), are reported in Table 9.

**Baselines.** The idea of transferring the knowledge from a pre-trained network to a deeper network before continuing training is similar to NET2NET. The term $M_B$ depth-aligned is referred to the model state after adding layers into the shallower model but before fusing with the deeper model. We use NET2NET to generate a RESNET34 model ($M_B$ depth-aligned) which preserves the performance of $M_B$. Note that $M_B$ depth-aligned has the same accuracy as the smaller model and the same architecture as the fused model. To compare with our method, we also finetune the pre-trained models, $M_B$ depth-aligned, and train a RESNET34 from scratch. As a reference, we report the result of ensemble learning that calculates the average predictions over all individual models.

**Results.** As can be seen from Table 1 retraining from the fused model gains an accuracy of 93.60, which is the highest among all initializations. This demonstrates the advantage of CLAFusion over NET2NET when initializing a large model from a pre-trained small model. Our method allows combining the knowledge from the pre-trained large model to generate a better initialization, rather than solely relying on the knowledge of the small model. Although the result is still lower than the accuracy of ensemble learning, the fused model almost halves the computational resources.

### 4.2.1 HETEROGENEOUS MODEL TRANSFER

**Settings.** Heterogeneous model transfer (Wang et al., 2021) is a branch of model transfer that deals with heterogeneous neural networks. It is contrasted to homogeneous model transfer, also known as transfer learning, in which the same network architecture is used in both the pre-training and finetuning phase. In this experiment, we apply CLA and OTFusion to the heterogeneous model transfer method (in short, HMT). We transfer the pre-trained RESNET18 to the pre-trained RESNET34 using different methods, then finetune and compare their performance. We use the same pre-trained models, layer representation, layer dissimilarity, and finetuning hyperparameters as in Section 4.2.

**Baselines.** The heterogeneous model transfer method (HMT) in the original paper (Wang et al., 2021) is used. Model parameters of the pre-trained RESNET18 are transferred to the pre-trained RESNET34. The weights that are not transferred are set to the weights of the pre-trained RESNET34. In the second method (HMT + Naive avg), after transferring we apply naive averaging to the transferred RESNET34 and the pre-trained RESNET34. Naive averaging is replaced with OTFusion in the third method (HMT + OTFusion). Different from the third method, the last method utilizes the optimal mapping from the CLA problem instead of the longest chain for layer-to-layer transfer.

**Results.** Table 2 demonstrates the average performance of different transfer methods. The second row reports the model accuracy before finetuning while the third row shows the final performance. Using CLA improves the performance of model transfer significantly. After finetuning, the combination of CLA and OTFusion leads to an improvement of 0.77 compared to using HMT only.

### 4.3 TEACHER-STUDENT FUSION

**Settings.** In this experiment, we transfer the knowledge from the pre-trained teacher model to the student model. We train two VGG models on CIFAR10 dataset: model A has VGG13 architecture

| # Params | Teacher | Students | | Finetune | | | |
|---|---|---|---|---|---|---|---|
| $(M_A, M_B, M_{\mathcal{F}})$ | $M_A$ | $M_B$ | $M_{\mathcal{F}}$ | $M_A$ | $M_B$ | $M_{\mathcal{F}}$ | $M_B$ depth-aligned |
| (33M, 3M, 3M) | 92.70 | 89.92 | 82.66 | 92.65 | 89.89 | **90.96** | 90.84 |

Table 3: The results of fusing teacher and student VGG models on CIFAR10 dataset . More results can be found in Table 13.

| | Distillation initialization | | | | |
|---|---|---|---|---|---|
| | Random $M_B$ | $M_B$ | $M_{\mathcal{F}}$ | $M_B$ depth-aligned | Random $M_{\mathcal{F}}$ |
| Best | 89.10 | 90.98 | **91.43** | 91.24 | 88.84 |
| Average | 88.41 ±0.77 | 90.73 ±0.43 | **91.16** ±0.25 | 91.09 ±0.19 | 88.10 ±0.68 |

Table 4: Knowledge distillation from teacher $M_A$ into student model $M_B$. Results in the last row are mean ±std. deviation over different temperatures. The detailed results are reported in Table 14.

with a double number of neurons at each layer while model B has VGG11 architecture with a half number of neurons at each layer. After fusing the teacher and student models, we retrain the fused model and compare it to the retraining result of the student model. The layer measure and the layer dissimilarity are identical to Section 4.2. We adopt the best hyperparameters as reported in (Singh & Jaggi, 2020, Section S11) for retraining. All finetuning hyperparameters are reported in Table 10.

**Results.** The classification accuracies are summarized in Table 3. After retraining, the fused model yields better performance than retraining other student models. The compression ratio between the fused model and the teacher is about 11 at a cost of reducing $1.74\%$ accuracy. This suggests that CLAFusion can act as a network pruning method to compress a heavy model into a lightweight one.

### 4.3.1 KNOWLEDGE DISTILLATION

**Settings.** As discussed in OTFusion, the fused model is an efficient initialization for knowledge distillation. To examine the same application of CLAFusion, we perform knowledge distillation to the above pre-trained VGG models. We employ the method in Hinton et al. (2015) to match the logit distribution of the student model to the teacher model. For initialization, we consider five different choices for the student model: (a) randomly initialized $M_B$, (b) $M_B$, (c) $M_{\mathcal{F}}$, (d) $M_B$ depth-aligned, (e) randomly initialized $M_{\mathcal{F}}$. We sweep over a set of hyperparameters to choose the best combination that maximizes the accuracy of the student model. The sets of hyperparameters for distillation also follows the setting in (Singh & Jaggi, 2020, Section S12): temperature $T = \{20, 10, 8, 4, 1\}$ and loss-weight factor $\gamma = \{0.05, 0.1, 0.5, 0.7, 0.95, 0.99\}$. For a fair comparison, the hyperparameters for training student models are identical to finetune hyperparameters in Section 4.3.

**Results.** The results for retraining and knowledge distillation are reported in Table 4, further results can be found in Table 14. The first row shows the best performance for all combinations of hyperparameters. The second row is the mean of the best accuracies obtained at different temperatures. Using the fused model as an initialization achieves the best performance among different choices of initializations. It showcases the application of CLAFusion as an efficient initialization for knowledge distillation. If averaging over different temperatures, retraining the fused model $(90.96)$ works better than vanilla distilling from the teacher model into the original student architecture $(88.41, 90.73)$. Even the best accuracy of vanilla distillation $(90.98)$ is slightly higher than that of retraining the fused model but comes with the cost of hyperparameter tuning. It further strengthens the claim that CLAFusion in combination with finetuning can act as a model compression method.

## 5 CONCLUSION

In the paper, we have presented a framework for model fusion. Our CLAFusion extends layer-wise model fusion methods to the setting of heterogeneous neural networks by solving a cross-layer alignment problem, followed by a layer balancing step. CLAFusion has been successfully applied in the setting of heterogeneous data as a skill transfer method. In addition, finetuning the fused network from CLAFusion achieves a better accuracy for RESNET trained on CIFAR10 dataset. Furthermore, it shows potential applications for model compression and knowledge distillation.

**Reproducibility Statement:** We provide our source codes for all experiments in the paper in the supplementary of the paper. The details of experimental settings, computational infrastructure, and other used public libraries are given in Appendix B. Neural network architectures and finetuning process of all experiments in the main text can be found in Appendix D. We also provide links to pre-trained models for each experiment in the README file of our source code.

**Ethics Statement:** We expect that there are no negative social impacts from our work.

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

# Supplement to "Model Fusion of Heterogeneous Neural Networks via Cross-Layer Alignment"

In this supplementary material, we first present some thoughts on the application of CLAFusion to multiple models scenario in Appendix A. The common settings, used assets, and computational resources are described in Appendix B. Next, we study the importance of constraints on the first and last layers as well as the efficiency of CLA in Appendix C. Finally, we provide additional results of our experiments in Appendix D.

## A  CLAFUSION FOR MULTIPLE NEURAL NETWORKS.

The cross-layer alignment problem for multiple neural networks is a non-trivial task. It can be formulated as a multi-index assignment problem (Spieksma, 2000; Huang & Lim, 2003). In addition, it has some connections to the multi-marginal optimal partial optimal transport (Figalli, 2010; Kitagawa & Pass, 2015). It is an interesting direction for future work. Nevertheless, there are two naive approaches to extend CLAFusion to the case of multiple networks. Consider K pre-trained models $\{M_i\}_{k=1}^K$. In the first approach, we can apply CLAFusion for two models iteratively $K-1$ times to fuse $K$ models. In each iteration, we fuse a pair of models and replace them with their fused model, thus the number of models reduces by 1. The second approach is similar to what they did in OTFusion (Singh & Jaggi, 2020). Starting with an estimation of the fused model $M_{\mathcal{F}}$. We apply the first and second parts of CLAFusion $K$ times to align $K$ pre-trained models with respect to the fused model. Finally, we average the weights of those aligned networks to produce the final weights for the fused model. The choice of $M_{\mathcal{F}}$ plays an important role in this approach. However, it is unclear how it is chosen in OTFusion.

**Results.** We tried two naive approaches to fuse multiple networks. Unfortunately, we could not achieve a desirable outcome. As for the first approach, it cumulatively deteriorates the performance of the fused networks after each iteration. For the second approach, we could not achieve a good initialization for the fused model as in OTFusion. It is important to stress that our settings are more difficult than previous model fusion methods because we consider two networks with different numbers of layers.

## B  EXPERIMENT SETTINGS

Following OTFusion, the layer-wise model fusion is computed using the activation-based alignment strategy in which the pre-activation matrix instead of the activation matrix is used for the neuron measure. The hyperparameters for pre-trained models are summarized in Table 5. Without any further specification, the random seed for training or retraining is set to its default value. The accuracy of pre-trained MLP is the accuracy of the last epoch. While the accuracies of VGG and RESNET are reported as the best performing checkpoint. For all finetuning experiments, we always chose the best record among all epochs.

|  | MLP | VGG | ResNet |
|---|---|---|---|
| Number of epochs | 10 | 300 | 300 |
| Training batch size | 64 | 128 | 256 |
| Test batch size | 1000 | 1000 | 1000 |
| Optimizer | SGD | SGD | SGD |
| Initial LR | 0.01 | 0.05 | 0.1 |
| Momentum | 0.5 | 0.9 | 0.9 |
| Weight decay |  | 0.0005 | 0.0001 |
| LR decay factor |  | 2 | 10 |
| LR decay epochs |  | 30,60,...,270 | 150,250 |
| Default seed | 0 | 42 | 42 |
| 5 seeds | 0,1,2,3,4 | 40,41,42,43,44 | 40,41,42,43,44 |

Table 5: The hyperparameters for training different network architectures.

| Pair | Layer representation | Cost function | Both | Last layer constraint | First layer constraint | None |
|------|---------------------|---------------|------|----------------------|------------------------|------|
| 1 | Number of neurons | squared Euclidean | [1, 5] | [1, 5] | [1, 3] | [1, 3] |
| | Pre-activation matrix | 1 - linear CKA | [1, 5] | [1, 5] | [1, 2] | [1, 2] |
| | Pre-activation matrix | Wasserstein distance | [1, 5] | [2, 5] | [1, 3] | [2, 3] |
| 2 | Number of neurons | squared Euclidean | [1, 2, 5] | [1, 2, 5] | [1, 2, 3] | [1, 2, 3] |
| | Pre-activation matrix | 1 - linear CKA | [1, 2, 5] | [1, 2, 5] | [1, 2, 3] | [1, 2, 3] |
| | Pre-activation matrix | Wasserstein distance | [1, 2, 5] | [1, 2, 5] | [1, 2, 3] | [1, 2, 3] |

Table 6: The optimal CLA for different combinations of layer representation and cost function.

**Open source code.** We adapt the official implementation of OTFusion[1] (Singh & Jaggi, 2020) in our implementation. For computing the optimal transport, we use Python Optimal Transport (POT) library[2] (Flamary et al., 2021). For model transfer, the source code of the original paper[3] (Wang et al., 2021) is utilized.

**Computational resources.** All experiments are done on 1 Tesla V100 GPU.

## C ABLATION STUDIES

**Settings.** We compare the performance of different combinations of layer representation and layer balancing methods in our framework. To prove the efficiency of the CLA step, we compare the result of fusing two models using different mappings. We also consider removing the constraint on the first and last hidden layers. We train two pairs of MLP in 5 different seeds. In both two pairs, model A has 5 hidden layers of size 400, 200, 100, 50, 25. Model B has 2 hidden layers of size 400, 100 in the first pair while it has 3 hidden layers of size 400, 200, 100 in the second pair.

**Results of the optimal mappings.** The cross-layer mappings obtained using Algorithm 1 with different combinations are given in Table 6. According to Definition 1 in Section 3.1, the optimal mapping is strictly increasing mapping $a : [\![n]\!] \mapsto [\![m]\!]$ where $n$ and $m$ are the number of layers of the shallower and deeper networks, respectively. In this case, we have $m = 5$, $n = 2$ for the first pair, and $n = 3$ for the second pair. The optimal mapping $a$ is represented as $[a(1), a(2), \ldots, a(n)]$. For example, [1, 2, 5] means that $a(1) = 1, a(2) = 2$, and $a(3) = 5$. We observe that changing the random seed barely affects the result of CLA in this setting. When keeping both first and last layer constraints, all three combinations yield the same mappings in both pairs. For the second pair, all three combinations share the same mappings in 4 out of 5 seeds even if none of the two constraints are placed (When the random seed is 2, the optimal mapping for pre-activation matrix + 1 - linear CKA is [1, 2, 5] instead of [1, 2, 3]). On the other hand, removing both constraints in the first pair makes each combination have a different mapping.

**The effectiveness of the optimal mapping.** We fuse each pair of MLPs for all 10 possible mappings between two models A and B. For the layer balancing method, we try both adding layers and merging layers (the sign of sum estimation). The average performance of pre-trained models and the fused model across 5 random seeds are summarized in Table 7. In three out of four scenarios, the best accuracy of the fused model is achieved when imposing both the first and last layer constraints. Removing the last layer constraint decreases the performance of the fused model. When removing the first layer constraint, the accuracy drops even more significantly. Because an input image may consist of negative cell values due to normalization, adding a new layer as the first hidden layer does not maintain the accuracy of the shallower model. Therefore, it is necessary to match the first and last hidden layers of the two models. In addition, the mappings obtained from the CLA step generally result in higher accuracy than other mappings, proving the efficiency of the CLA. Comparing between layer balancing methods, the merging layers method runs slower but leads to a higher accuracy than the adding layers in both pairs.

---

[1] https://github.com/sidak/otfusion

[2] https://github.com/PythonOT/POT

[3] https://anonymous.4open.science/r/6ab184dc-3c64-4fdd-ba6d-1e5097623dfd/a_hetero_model_transfer.py

| Pair | Mapping | $M_A$ | $M_B$ | $M_{\mathcal{F}}$ | |
|------|---------|-------|-------|-----|-----|
| | | | | Add | Merge |
| 1 | [1, 2] | 96.95 | 97.59 | 91.91 | **95.11** |
| | [1, 3] | | | 91.95 | 94.20 |
| | [1, 4] | | | 92.26 | 93.30 |
| | [1, 5] | | | **92.38** | 93.18 |
| | [2, 3] | | | 79.20 | 62.93 |
| | [2, 4] | | | 81.73 | 57.09 |
| | [2, 5] | | | 81.05 | 58.17 |
| | [3, 4] | | | 75.42 | 56.48 |
| | [3, 5] | | | 75.28 | 58.34 |
| | [4, 5] | | | 72.68 | 56.00 |
| 2 | [1, 2, 3] | 96.95 | 97.75 | 92.33 | 93.90 |
| | [1, 2, 4] | | | 92.48 | 93.41 |
| | [1, 2, 5] | | | **92.49** | **94.00** |
| | [1, 3, 4] | | | 90.69 | 92.21 |
| | [1, 3, 5] | | | 90.88 | 93.04 |
| | [1, 4, 5] | | | 90.98 | 93.06 |
| | [2, 3, 4] | | | 82.21 | 51.26 |
| | [2, 3, 5] | | | 83.14 | 56.56 |
| | [2, 4, 5] | | | 83.00 | 54.16 |
| | [3, 4, 5] | | | 79.49 | 56.31 |

Table 7: Performance comparison between different combinations of mapping and balancing methods.

| | Seed | $w_B$ | | | | | | | | | | |
|---|------|-----|-----|-----|-----|-----|-----|-----|-----|-----|-----|-----|
| | | 0.0 | 0.1 | 0.2 | 0.3 | 0.4 | 0.5 | 0.6 | 0.7 | 0.8 | 0.9 | 1.0 |
| Adding layers | 0 | 86.10 | 87.06 | **88.23** | 78.64 | 73.48 | 69.70 | 70.45 | 75.91 | 81.38 | 85.36 | 87.68 |
| | 1 | 91.60 | **91.81** | 91.23 | 85.63 | 77.20 | 77.99 | 81.59 | 84.85 | 86.88 | 87.48 | 87.29 |
| | 2 | 90.80 | 92.07 | **92.46** | 84.06 | 82.53 | 83.21 | 85.01 | 86.83 | 87.76 | 88.01 | 87.83 |
| | 3 | 91.94 | **92.31** | 91.39 | 79.58 | 71.85 | 71.77 | 78.31 | 83.63 | 85.79 | 87.01 | 87.51 |
| | 4 | **93.34** | 93.31 | 91.56 | 81.70 | 77.45 | 78.46 | 82.15 | 85.00 | 86.62 | 87.37 | 87.58 |
| | Avg | 90.76 | **91.31** | 90.97 | 81.92 | 76.50 | 76.23 | 79.50 | 83.24 | 85.69 | 87.05 | 87.58 |
| Merging layers | 0 | 86.10 | **87.19** | 87.10 | 78.51 | 63.66 | 56.74 | 53.31 | 53.88 | 57.65 | 66.55 | 76.35 |
| | 1 | 91.60 | **92.23** | 91.83 | 89.08 | 78.69 | 76.04 | 77.34 | 81.44 | 84.87 | 86.39 | 86.60 |
| | 2 | 90.80 | 91.98 | **92.30** | 86.00 | 81.27 | 79.42 | 80.09 | 82.95 | 85.46 | 85.61 | 85.04 |
| | 3 | 91.94 | 92.66 | **92.76** | 88.13 | 71.65 | 65.90 | 64.20 | 67.39 | 72.91 | 76.06 | 77.40 |
| | 4 | 93.34 | **93.80** | 93.18 | 90.03 | 80.24 | 79.27 | 81.01 | 82.52 | 83.91 | 84.68 | 85.09 |
| | Avg | 90.76 | **91.57** | 91.43 | 86.35 | 75.10 | 71.47 | 71.19 | 73.64 | 76.96 | 79.86 | 82.10 |

Table 8: Performance comparison between two layer balancing methods on skill transfer task. Experiments are run 5 times.

# D ADDITIONAL EXPERIMENT RESULTS

## D.1 SKILL TRANSFER

**Results.** The results of skill transfer for 5 different random seeds are reported in Table 8. The best accuracy for each seed is written in bold font. The special list model A generally has higher accuracy than the generalist model B seeds because it has been trained on all 10 labels. In 4 out of 5 seeds, the fused model improves over both individual models when the weight proportion of model B is small ($w_B = \{0.1, 0.2\}$).

**Comparison between two layer balancing methods.** For comparison purposes, we further conduct the same procedure but replace adding layers by merging layers (the sign of sum estimation). Similar to the previous experiment, the fused model attains the best performance when the weight proportion of model B is small ($w_B = \{0.1, 0.2\}$). Figure 5 illustrates the average performance of two layer balancing methods on the skill transfer task. When averaging over 5 random seeds, both methods

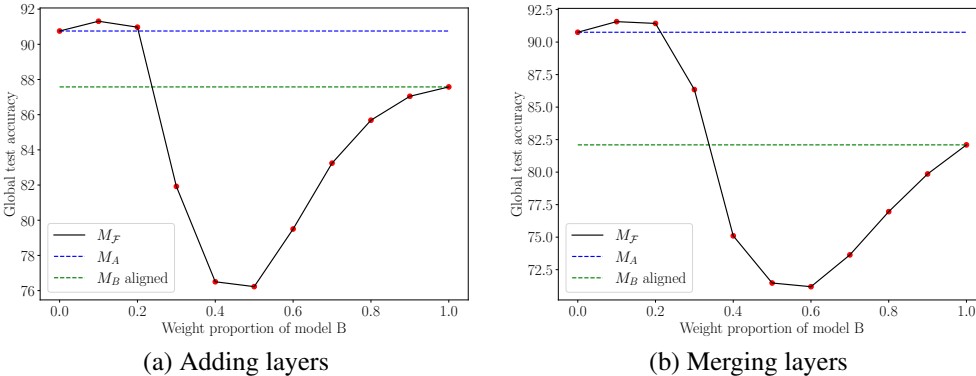

Figure 5: The average performance of skill transfer using (a) adding layers and (b) merging layers as the layer balancing method.

| Number of epochs | 120 |
|---|---|
| Training batch size | 256 |
| Test batch size | 1000 |
| Optimizer | SGD |
| Initial LR | 0.1 |
| Momentum | 0.9 |
| Weight decay | 0.0001 |
| LR decay factor | 2 |
| LR decay epochs | 20,40,60,80,100 |

Table 9: Finetuning RESNET hyperparameters.

| Number of epochs | 120 |
|---|---|
| Training batch size | 128 |
| Test batch size | 1000 |
| Optimizer | SGD |
| Initial LR | 0.01 |
| Momentum | 0.9 |
| Weight decay | 0.0005 |
| LR decay factor | |
| LR decay epochs | 20,40,60,80,100 |

Table 10: Finetuning teacher-student models hyperparameters.

| Seed | $M_A$ | $M_B$ | Ensemble | $M_{\mathcal{F}}$ | | | | Finetune | |
|---|---|---|---|---|---|---|---|---|---|
| | | | Learning | | $M_A$ | $M_B$ | $M_{\mathcal{F}}$ | $M_B$ depth-aligned | Random RESNET34 |
| 40 | 93.42 | 93.21 | 94.10 | 67.83 | 93.50 | 93.47 | **93.60** | 93.25 | 91.95 |
| 41 | 93.37 | 92.90 | 93.81 | 63.16 | 93.61 | 93.26 | **93.73** | 93.43 | 91.93 |
| 42 | 93.31 | 93.05 | 93.94 | 68.44 | 93.61 | 93.36 | **93.77** | 93.19 | 92.25 |
| 43 | 93.39 | 92.78 | 93.71 | 66.60 | 93.50 | 93.30 | 93.47 | 93.20 | 91.76 |
| 44 | 93.06 | 92.65 | 93.49 | 61.86 | 93.37 | 93.05 | **93.41** | 93.01 | 92.12 |
| Avg | 93.31 | 92.92 | 93.81 | 65.58 | 93.52 | 93.29 | **93.60** | 93.22 | 92.00 |

Table 11: The results of finetuning RESNET from different choices of initialization across 5 different seeds.

perform best if $w_B = 0.1$. The best accuracy of merging layers (91.57) is slightly higher than that of adding layers (91.31) even though the accuracy of $M_B$ depth-aligned drops from 87.58 to 82.10.

## D.2    GENERATE AN EFFICIENT INITIALIZATION

**Results.** Finetuning from the fused model always improves the accuracy of the pre-trained models as illustrated in Table 11. In comparison with other initializations, it yields the best performance in 4 out of 5 seeds. $M_B$ depth-aligned, which is the result of NET2NET (Chen et al., 2015) operation, only performs better than finetuning from scratch and be comparable to that of HMT. Starting from the results of NET2NET and HMT are even worse than that from the smaller pre-trained model. (Note that $M_B \rightarrow M_A$ divergences with $LR = 0.1$ so its accuracy is 10, which equals to random guessing.)

| Method | Seed | HMT | HMT + Naive avg | HMT + OTFusion | HMT + CLA + OTFusion |
|--------|------|-----|-----------------|----------------|----------------------|
| Transfer | 40 | 9.55 | 37.80 | 20.17 | **60.72** |
| | 41 | 10.13 | 37.66 | 17.66 | **52.09** |
| | 42 | 10.45 | 43.44 | 19.25 | **55.94** |
| | 43 | 12.44 | 39.24 | 15.20 | **55.82** |
| | 44 | 11.37 | 29.33 | 19.30 | **51.55** |
| | Avg | 10.79 | 37.49 | 18.32 | **55.22** |
| Finetune | 40 | 93.29 | 93.63 | 93.61 | **93.76** |
| | 41 | 93.14 | 93.28 | **93.69** | **93.69** |
| | 42 | 92.48 | 93.16 | **93.78** | 93.68 |
| | 43 | 93.20 | 93.27 | **93.80** | 93.62 |
| | 44 | 92.93 | 83.35 | 93.41 | **93.75** |
| | Avg | 92.93 | 91.34 | 93.66 | **93.70** |

Table 12: The results of model transfer from $M_B$ to $M_A$ across 5 different seeds.

| Seed | $M_A$ | $M_B$ | Ensemble Learning | $M_{\mathcal{F}}$ | $M_A$ | $M_B$ | $M_{\mathcal{F}}$ | $M_B$ depth-aligned |
|------|-------|-------|-------------------|-------------------|-------|-------|-------------------|---------------------|
| 40 | 92.71 | 89.68 | 92.53 | 81.66 | 92.67 | 89.81 | **90.64** | 90.54 |
| 41 | 92.34 | 89.61 | 92.68 | 82.59 | 92.50 | 89.57 | **90.59** | 90.42 |
| 42 | 92.70 | 89.92 | 92.86 | 82.66 | 92.65 | 89.89 | **90.96** | 90.84 |
| 43 | 92.79 | 89.78 | 92.82 | 81.79 | 92.77 | 90.10 | 90.55 | **90.61** |
| 44 | 92.70 | 89.62 | 92.51 | 81.43 | 92.56 | 89.46 | **90.44** | 90.19 |
| Avg | 92.65 | 89.72 | 92.68 | 82.03 | 92.63 | 89.77 | **90.64** | 90.52 |

The "Retrain" columns ($M_A$, $M_B$, $M_{\mathcal{F}}$, $M_B$ depth-aligned) are grouped under a shared "Retrain" header.

Table 13: Finetuning teacher-student VGG across 5 different seeds. The results reported in Table 3 are run with the random seed of 42.

### D.3 HETEROGENEOUS MODEL TRANSFER

**Results.** Table 11 details the results of 4 different model transfer methods. Combining CLA and OTFusion with HMT always results in a great boost (at least 17.73) in the accuracy of the transferred model. After finetuning, it gives the best performance in 3 out of 5 random seeds.

### D.4 TEACHER-STUDENT FUSION

**Results.** Table 13 illustrates the finetuning result of teacher and student models across 5 different seeds. The fused model $M_{\mathcal{F}}$ is the most productive initialization of student models in 4 out of 5 seeds. Retraining the fused model always yields higher accuracy than continuing training the student model.

**Discussion.** At first glance, our approach may resemble the NET2NET operations. Although both approaches increase the depth of the student network and use the deeper student as a good initialization, there are two major differences between ours and NET2NET. Firstly, we present a systematic way of finding the location to add the identity mappings for different types of network architectures while NET2NET is manually designed for a specific type of network. Secondly, we do not enlarge the width of the student network (equivalently, use only NET2DEEPNET operation) so that the student network remains more compact than the teacher network.

### D.5 KNOWLEDGE DISTILLATION

**Results.** The results of distilling the teacher model into the student model are given in Table 14. Knowledge distillation from the fused model leads to the best accuracy in 4 out of 5 temperatures. In addition, both its average and best performance are the highest among all initializations. $M_B$ depth-aligned, which is obtained from the NET2NET operation, is the second-best choice, followed by the pre-trained student model $M_B$. Naive knowledge distillation into either random initialized student models, however, yields a much lower accuracy. This suggests that CLAFusion can serve as an efficient initialization for knowledge distillation.

| Temperature | Distillation initialization | | | | |
|---|---|---|---|---|---|
| T | Random $M_B$ | $M_B$ | $M_\mathcal{F}$ | $M_B$ depth-aligned | Random $M_\mathcal{F}$ |
| 20 | 88.99 (0.70) | 90.94 (0.70) | **91.29** (0.10) | 91.14 (0.70) | 88.20 (0.10) |
| 10 | 89.10 (0.70) | 90.97 (0.70) | **91.23** (0.50) | **91.23** (0.10) | 88.84 (0.50) |
| 8 | 88.91 (0.70) | 90.98 (0.70) | **91.43** (0.70) | 91.24 (0.70) | 88.76 (0.50) |
| 4 | 87.89 (0.70) | 90.81 (0.99) | **91.14** (0.50) | 91.10 (0.50) | 87.65 (0.05) |
| 1 | 87.14 (0.10) | 89.97 (0.10) | 90.69 (0.05) | **90.73** (0.05) | 87.04 (0.05) |
| Best | 89.10 | 90.98 | **91.43** | 91.24 | 88.84 |
| Avg | 88.41 | 90.73 | **91.16** | 91.09 | 88.10 |

Table 14: Distillation results for different temperatures. Each entity is the best accuracy obtained by varying the loss-weight factor. The last two rows report the best and average performance across 5 different temperatures.

