# OpenReview forum: "Model Fusion of Heterogeneous Neural Networks via Cross-Layer Alignment"
_ICLR.cc/2022/Conference — ICLR 2022 Submitted_

### Official Review · Reviewer_1twh · 2021-10-27

**Correctness:** 3
**Technical Novelty And Significance:** 2
**Empirical Novelty And Significance:** 2
**Recommendation:** 3
**Confidence:** 4

**Main Review:**

This paper presents an algorithm named CLAFusion. The goal is to fuse two networks with different architectures so that the fused one has reasonable computational complexity and works favorably compared to the separate ones. Experiments on CIFAR10 and small neural networks show the effectiveness.

I have the following concerns about this paper.

First, I am thinking of the practical value of this algorithm, as well as the OTFusion algorithm. If the cost of training multiple to-be-fused networks is acceptable, the baseline that simply uses all these costs to train the target network should be compared. For example, if training a ResNet34 needs 10 hours and a ResNet18 needs 6 hours, then what is the performance of training a ResNet34 for 16 hours? Many prior works showed that simply extending the training epochs can improve classification accuracy.

Second, the experimental part only shows CIFAR10 results. This is far below expectation. Since training-from-scratch is not required, I would expect ImageNet results to be offered, or at least CIFAR100 can be tested. We know that CIFAR10 is a very small and relatively simple dataset, and more importantly, the classification accuracy on this dataset often suffers ~0.5% random noise, so using it as the single testbed is not acceptable to me. BTW, the deviation of different algorithms should be reported, especially when the gain is smaller than 0.5% (e.g. Table 1).

Third, I tried to read the paper multiple times, but I cannot find the technical description that how two models with the same number of layers are combined. Section 3 ends with adding layers to the shallow network or merging layers in the deep network, but I cannot find the next procedure. Is it missing in the paper?

BTW, I cannot say that the paper is well organized. It spent too much space in the introduction and background parts, but many details (e.g. how to compute the cost matrix or how to combine the networks) are not discussed.

**Summary Of The Paper:**

This paper presents an algorithm named CLAFusion. The goal is to fuse two networks with different architectures so that the fused one has reasonable computational complexity and works favorably compared to the separate ones. Experiments on CIFAR10 and small neural networks show the effectiveness.

**Summary Of The Review:**

1. The practical value is questionable.
2. The description of the approach is not complete.
3. Experiments are only performed on very small datasets.

Overall, this paper is not well prepared for publication.

---

> ### Author Response · Authors · 2021-11-17
> **Response to Reviewer 1twh**
>
> **Q1**: First, I am thinking of the practical value of this algorithm, as well as the OTFusion algorithm. If the cost of training multiple to-be-fused networks is acceptable, the baseline that simply uses all these costs to train the target network should be compared. For example, if training a ResNet34 needs 10 hours and a ResNet18 needs 6 hours, then what is the performance of training a ResNet34 for 16 hours? Many prior works showed that simply extending the training epochs can improve classification accuracy.
>
> **Answer**: Thanks for your interesting question. Both OTFusion and our work considered the extending training scenario. When we trained the fused model for 200 epochs, we also continued training the original networks for another 200 epochs and reported their performance in Tables 1, 3, and 4.
>
> **Q2**: Second, the experimental part only shows CIFAR10 results. This is far below expectation. Since training-from-scratch is not required, I would expect ImageNet results to be offered, or at least CIFAR100 can be tested. We know that CIFAR10 is a very small and relatively simple dataset, and more importantly, the classification accuracy on this dataset often suffers ~0.5% random noise, so using it as the single testbed is not acceptable to me. BTW, the deviation of different algorithms should be reported, especially when the gain is smaller than 0.5% (e.g. Table 1).
>
> **Answer**: Thanks for your good comment. We run each experiment for 5 different seeds then reported the averaging performance in the main text. The detailed results can be found in Appendix D. Actually we had to train pre-trained models from scratch for all experiments. In addition, almost all of the experiments require an extra retraining process and we also run each experiment 5 times. Therefore, we cannot offer the results for big datasets at the moment.
>
> **Q3**: Third, I tried to read the paper multiple times, but I cannot find the technical description that how two models with the same number of layers are combined. Section 3 ends with adding layers to the shallow network or merging layers in the deep network, but I cannot find the next procedure. Is it missing in the paper?
>
> **Answer**: Thanks for your interesting question. In this paper, we did not propose a new method to combine two models with the same number of layers. Because any available layer-wise model fusion methods can be incorporated into our framework to handle this task. In our experiments, we adopted OTFusion as the third component of our framework.
>
> **Q4**: BTW, I cannot say that the paper is well organized. It spent too much space in the introduction and background parts, but many details (e.g. how to compute the cost matrix or how to combine the networks) are not discussed.
>
> **Answer**: Thanks for your helpful suggestion. We shortened the introduction and background sections and brought more details of the framework and experiments from the Appendix into the main text. Please check our latest version.

---

### Official Review · Reviewer_pWLE · 2021-11-03

**Correctness:** 3
**Technical Novelty And Significance:** 2
**Empirical Novelty And Significance:** 2
**Recommendation:** 5
**Confidence:** 4

**Main Review:**

I have two main concerns about this paper.
(1)Neural networks have many layers and neural networks are non-linear. It is unclear that simply computing average models over two models ($A$ and $B^{‘}$) leads to better performance compared with $A$. For example, for layer $l$, $A$ and $B$ have weights $W_A^l$ and $W_{B^{‘}}^l$, the input of this layer is denoted as $x$, we obtain $W_A^lx$ and $W_{B^{‘}}^lx$. It is hard to understand to fuse $f(W_A^lx)$ and $f(W_{B^{‘}}^lx)$ as $f((W_A^l+W_{B^{‘}}^l)x)$ where $f(\cdot)$ is a non-linear activation function.

(2)The experiments are very weak. For Table 1, $M_A$ obtains $93.52\%$ and the $M_F$ obtains $93.60\%$. For Table 2, HMT+CLA+OTFUSION obtains $93.70\%$ against $93.52\%$. For Tables 3 and 4, $M_F$ obtains about $1.07\%$ and $0.43\%$ improvement, respectively. But only VGGNet is provided. Extending the experiments to state-of-the-art networks and larger datasets are suggested.

(3) For Table 3 and Table 4, why does $M_F$ have 3M parameters as $M_B$, and $M_A$ have 33 M parameters. Do the authors use merging layers here? As mentioned before, the authors claim that the same merging method does not work in the case of CNN.

**Summary Of The Paper:**

This paper proposes a layer-wise model fusion method that fuses neural networks with a different number of layers by cross-layer alignment. Given two networks $A$ and $B$ ($A$ is deeper than $B$), the framework first expands $B$ as $B^{’}$ via cross-alignment such that the structure of $B^{‘}$ is the same as that of $A$ while keeping mapping unchanged. Then it performs fusion for $B^{‘}$ and $A$. The idea is interesting. The experiments show slight improvements compared with baselines.

**Summary Of The Review:**

(1) Model fusion is a challenging problem in deep learning. The topic is interesting.
(2) The explanations and analyses of the proposed model fusion are not convincing.
(3) More expensive experiments are needed.

---

> ### Author Response · Authors · 2021-11-17
> **Response to Reviewer pWLE: Part 1**
>
> **Q1**: Neural networks have many layers and neural networks are non-linear. It is unclear that simply computing average models over two models (A and B) leads to better performance compared with A. For example, for layer l, A and B have weights $W_A^l$ and $W_B^l$, the input of this layer is denoted as x, we obtain $W_A^l x$ and $W_B^l x$.It is hard to understand to fuse $f(W_A^l x)$ and $f(W_B^l x)$ as $f((W_A^l + W_B^l)x)$ where $f(\cdot)$ is a non-linear activation function.
>
> **Answer**:
> Thanks for your interesting question. First, let us state some important points for clarification. It is actually $f(\frac{W_A^l + W_B^l}{2} x)$, which is the weight averaging mechanism. Similar to recent work in the literature, we did not immediately apply weight averaging but performed an alignment beforehand. The fused model does not always lead to better performance. It only happens in the skill transfer experiment on a synthetic dataset that we adopted from the OTFusion paper. Generally, the fused model serves as a good initialization which requires an extra retraining process to improve the performance over the original pre-trained networks. The same scheme of model fusion + retraining is widely used in related works [R.1, R.2, R.3, R.4]. One intuition why weight averaging may work is that averaging model parameters yields a good estimator (reducing variance) of the optimal weight [R.5]. Another intuition is that vanilla averaging resembles ensemble learning which works surprisingly well in practice. Our settings consider two feed-forward networks using ReLU activation in each layer. Assume that $W_A$ and $W_B$ are relatively close in the weight space (after some operations), then
> $$
> \widehat{\boldsymbol{Y}}
> = \text{ReLU}(\frac{1}{2}\boldsymbol{W}_A \boldsymbol{X} + \frac{1}{2} \boldsymbol{W}_B \boldsymbol{X})
> \approx \text{ReLU}(\frac{1}{2} \boldsymbol{W}_A \boldsymbol{X}) + \text{ReLU}(\frac{1}{2} \boldsymbol{W}_B \boldsymbol{X})
> = \frac{1}{2}(\boldsymbol{Y}_1 + \\boldsymbol{Y}_2)
> $$.
>
> **Q2**: The experiments are very weak. For Table 1, $M_A$ obtains $93.52$ and the $M_F$ obtains $93.60$. For Table 2, HMT+CLA+OTFUSION obtains $93.70$ against $93.52$. For Tables 3 and 4, $M_F$ obtains about $1.70$ and $0.43$ improvement, respectively. But only VGGNet is provided. Extending the experiments to state-of-the-art networks and larger datasets are suggested.
>
> **Answer**: Thanks for your constructive suggestion. Due to the combinatorial hardness of the model fusion problem, the fused model did not achieve a large margin compared to pre-trained networks. The same phenomena were also observed in the experiments of OTFusion. We incorporated OTFusion into the CLAFusion framework so the problem became worse. Although our framework can adopt other layer-wise model fusion methods in the third step, it is not the principal purpose of our paper.
>
> In essence, recent methods [R.1, R.2, R.3, R.4] in the model fusion literature are still based on the weight averaging mechanism. However, as you pointed out, the application of model fusion is still limited to some specific architectures (VGG, ResNet), leaving future directions for complicated architectures such as LSTM, Transformer, etc. The main reason is due to a lack of an appropriate to combine weights of complex components such as LSTM cell or attention module.
>
> References:
>
> [R.1] Mikhail Iu. Leontev, Viktoriia Islenteva, and Sergey V. Sukhov. Non-iterative knowledge fusion in deep convolutional neural networks. Neural Processing Letters, 51(1):1–22, Jul 2019.
>
> [R.2] Sidak Pal Singh and Martin Jaggi. Model fusion via optimal transport. Advances in Neural Information Processing Systems, 33, 2020.
>
> [R.3] Joshua Smith and Michael Gashler. An investigation of how neural networks learn from the experiences of peers through periodic weight averaging. In 2017 16th IEEE International Conference on Machine Learning and Applications (ICMLA), pages 731–736. IEEE, 2017.
>
> [R.4] Hongyi Wang, Mikhail Yurochkin, Yuekai Sun, Dimitris Papailiopoulos, and Yasaman Khazaeni. Federated learning with matched averaging. arXiv preprint arXiv:2002.06440, 2020.
>
> [R.5] Joachim Utans. Weight averaging for neural networks and local resampling schemes. In Proc. AAAI-96 Workshop on Integrating Multiple Learned Models. AAAI Press, pages 133–138. Citeseer, 1996.

---

> ### Author Response · Authors · 2021-11-17
> **Response to Reviewer pWLE: Part 2**
>
> **Q3**: For Table 3 and Table 4, why does $M_F$ have 3M parameters as $M_B$, and $M_A$ have 33 M parameters. Do the authors use merging layers here? As mentioned before, the authors claim that the same merging method does not work in the case of CNN.
>
> **Answer**: Thanks for your interesting question. In Table 3, the size of $M_A, M_B$, and $M_F$ are 33M, 3M, and 3M, respectively. We still use adding layers on this occasion. Let us clarify the setting of this experiment which we initially put in Appendix D.4. Model A has VGG13 architecture with a double number of neurons at each layer while model B has VGG11 architecture with a half number of neurons at each layer. $M_F$ was extended to have the same number of layers as $M_A$ but its layer roughly has the same number of neurons as $M_B$. Therefore, the number of model parameters does not increase much compared to $M_B$. On the other hand, at each corresponding layer, the number of neurons of $M_F$ is only about $\frac{1}{4}$ that of $M_A$. Consequently, $M_F$ is much more compact than $M_A$.

---

### Official Review · Reviewer_LJwg · 2021-11-04

**Correctness:** 3
**Technical Novelty And Significance:** 2
**Empirical Novelty And Significance:** 3
**Recommendation:** 5
**Confidence:** 5

**Main Review:**

**Pros:**
- Relatively simple bag of tricks that are reasonably motivated and that enable OTFusion to handle different number of layers, which importantly reduces a key shortcoming of OTFusion.
- Empirical demonstration on several settings is thoroughly done and shows a decent advantage of the fusion approach.
- It seems that some ablations are also carried out for the particular choices made in their framework. E.g., enforcing the mapping of first and last hidden layers. But this is all in the supplementary and the description could be made more clear.

**Cons:**

*No demonstration for >2 networks:* The main drawback of this approach is that its empirical application is limited to the case of only 2 networks. It is not clear how components of their approach like Algorithm 1 will behave when applied to the case of > 2 networks. While the authors provide a descriptive account in the appendix of a possible way around, but the concrete application is still up in the air. It is important that authors provide at least some, perhaps preliminary, evidence in this regard. (This is unlike most related work on model fusion which shows a much thorough demonstration for multiple-model setting)

*Clarity, presentation, empirical details:* This is another very important aspect that the authors should really work hard upon, as currently, the paper is hard to read.
1. Many important empirical details are missing in the main text, and one has to go hunting in the appendix to understand what's going on. This is despite that the paper is not as space-constrained either. E.g., how two layers are adjudged to be similar, heterogenous model transfer, etc. Similarly, the table captions could be a bit more detailed.
2. Several references are made to OTFusion, but it gets clarified only when referenced on page 3. Please make this clear upfront.
3. The reference for model fusion on page 2 to (Claici et al, 2020) is inaccurate in this particular context. Please use something more appropriate like Wang et. al. 2020 and/or Singh & Jaggi 2020.
4. 'M_B aligned' is not as clear or even slightly misleading. Perhaps an alternative could be 'M_B depth-aligned'? This is because M_B is still not aligned with M_A in the sense of permutations.
5. This paper heavily relies on OTFusion, and thus I think perhaps a more informative abbreviation is  'CLOTFusion'.
6. I suppose that what you refer as 'HMT' is using Wang et. al., 2021 to first fuse Resnet18 and Resnet34 into a fused ResNet34 model 'HMT', which you then further combine with the original ResNet34 via naive avg, OTFusion, CLAFusion? Besides, what is the 'TRANSFER' in Table 2 and why does it have so dismal performance as compared to 'FINETUNE'?
7. Can you explain how Section 4.2 is inspired from Net2Net (whose reference, btw, is missing)? To me, it seems to be more similar to the setting in OTFusion of Tables 1-3.
8. Tables 6, 7 in the appendix are quite hard to read. Can you define and better explain the notational usage there?

*Language issues:* This is a minor but understandable issue. Yet, strictly speaking, on many occasions, it prevents a precise understanding of what is being articulated. Please have the text be proofread or use something like Grammarly. Some examples of weird/ungrammatical language constructs include: "still needs to be secure" -> "still needs to be secured", "stark development", "It was empirically found that vanilla averaging combines", "Because vanilla averaging did not study the permutation invariance nature". Besides, in many places, appropriate punctuation marks are missing.

I am happy to improve my score if the above concerns are adequately addressed.

**Summary Of The Paper:**

This work extends the OTFusion approach, for fusing multiple neural networks into a single neural network, to handle the case of networks with different number of layers. To this end, the authors first find a mapping between layers of the shallower network to that of the deeper network (by a dynamic programming approach, reminiscent of longest-subsequence like problems), then balance the number of layers (by either adding identity connections in one or merging layers), after which OTFusion is called upon to do the rest. The authors provide an empirical demonstration of their approach in similar settings as in the OTFusion paper, but where they can now successfully handle networks with different numbers of layers (but in the setting of two input networks).

**Summary Of The Review:**

The extension of model fusion (OTFusion) to heterogeneous settings (in the sense of different depths) seems to be well-executed. There is still room for many of the choices to be more streamlined or automated, but nevertheless does show fairly satisfactory results across various settings. The other important drawback is that the current empirical demonstration is limited to the case of 2 networks. The lack of clarity in the presentation, at some places, does not help either.

---

> ### Author Response · Authors · 2021-11-17
> **Response to Reviewer LJwg: Part 1**
>
> **No demonstration for >2 networks:**
>
> **Q1**: The main drawback of this approach is that its empirical application is limited to the case of only 2 networks. It is not clear how components of their approach like Algorithm 1 will behave when applied to the case of > 2 networks. While the authors provide a descriptive account in the appendix of a possible way around, but the concrete application is still up in the air. It is important that authors provide at least some, perhaps preliminary, evidence in this regard. (This is unlike most related work on model fusion which shows a much thorough demonstration for multiple-model setting)
>
> **Answer**: Thanks for your constructive suggestion. We have tried two naive approaches to fuse multiple networks. Unfortunately, we could not achieve a desirable outcome. As for the first approach, it cumulatively deteriorates the performance of the fused networks after each iteration. For the second approach, we could not achieve a good initialization for the fused model as in OTFusion. It is important to stress that our settings are more difficult than previous model fusion methods. We consider two networks with different numbers of layers.

---

> ### Author Response · Authors · 2021-11-17
> **Response to Reviewer LJwg: Part 2**
>
> **Clarity, presentation, empirical details**:
>
> **Q2**: Many important empirical details are missing in the main text, and one has to go hunting in the appendix to understand what's going on. This is despite that the paper is not as space-constrained either. E.g., how two layers are adjudged to be similar, heterogenous model transfer, etc. Similarly, the table captions could be a bit more detailed.
>
> **Answer**: Thanks for your constructive suggestion. We have modified our paper to bring experiment details including the settings and the description of baselines into the Experiments section. In addition, Algorithm 1 and the discussion about layer representation of CLA were also moved from the Appendix to the main text. Please check our revised version.
>
> **Q3**: Several references are made to OTFusion, but it gets clarified only when referenced on page 3. Please make this clear upfront.
>
> **Answer**: Thanks for your good suggestion. We brought the reference to OTFusion into the Introduction section. Please check our revised version.
>
> **Q4**: The reference for model fusion on page 2 to (Claici et al, 2020) is inaccurate in this particular context. Please use something more appropriate like Wang et. al. 2020 and/or Singh & Jaggi 2020.
>
> **Answer**: Thanks for your constructive suggestion. We were meant to use the definition of model fusion in their paper. We fixed the citation in our latest version.
>
> **Q5**:  'M_B aligned' is not as clear or even slightly misleading. Perhaps an alternative could be 'M_B depth-aligned'? This is because M_B is still not aligned with M_A in the sense of permutations.
>
> **Answer**: Thanks for your insightful comment. We referred to the state of model B which is still not permuted to match the neuron orders of model A yet. Therefore, M_B depth-aligned is a more appropriate term. We fixed it in our latest version.
>
> **Q6**: This paper heavily relies on OTFusion, and thus I think perhaps a more informative abbreviation is 'CLOTFusion'.
>
> **Answer**: Thanks for your interesting comment. The results for our experiments were indeed based on OTFusion. However, in this paper what we want to propose is the general framework, not a particular method. We used OTFusion as a component in our framework because it is the most suitable one. Nevertheless, our framework CLAFusion can adopt any other layer-wise model fusion method.
>
> **Q7**: I suppose that what you refer as 'HMT' is using Wang et. al., 2021 to first fuse Resnet18 and Resnet34 into a fused ResNet34 model 'HMT', which you then further combine with the original ResNet34 via naive avg, OTFusion, CLAFusion? Besides, what is the 'TRANSFER' in Table 2 and why does it have so dismal performance as compared to 'FINETUNE'?
>
> **Answer**: Thanks for your insightful question. Your interpretation is correct. Originally, we described competing methods in Appendix D.3. We brought it to the main text in our latest revision. ‘TRANSFER’ is the model accuracy before finetuning. As shown in the table, simply applying heterogeneous model transfer methods did not yield a favorable model but a good initialization, which was also observed in their original paper. An extra retraining step was needed to produce a good model. The final performance was reported in the ‘FINETUNE’ row.
>
> **Q8**: Can you explain how Section 4.2 is inspired from Net2Net (whose reference, btw, is missing)? To me, it seems to be more similar to the setting in OTFusion of Tables 1-3.
>
> **Answer**: Thanks for your constructive comment. We here consider an application of CLAFusion which is similar to Net2Net in the following sense. In Net2Net, the authors transfer the knowledge from a pre-trained small network into a large randomly initialized network. In Section 4.2, CLAFusion transfers knowledge from a pre-trained ResNet18 into a pre-trained (instead of randomly initialized) ResNet34. Both methods add new layers into the smaller network while preserving the information. On the other hand, the setting is also similar to the setting in Section 5.3 of OTFusion as you mentioned. The main difference is that we fuse two networks having different numbers of layers. We have rewritten this part so please check our revised version.
>
> **Q9**: Tables 6, 7 in the appendix are quite hard to read. Can you define and better explain the notational usage there?
>
> **Answer**: Thank you for your suggestion. According to Definition 1 in Section 3.1, the optimal mapping is strictly increasing mapping from $a : [1, 2, \ldots, n] \mapsto [1, 2, \ldots, m]$ where n and m are the number of layers of the shallower and deeper networks, respectively. In this case, we have $m = 5, n = 2$ or $3$. The optimal mapping $a$ is represented as $[a(1), a(2), \ldots, a(n)]$ in Tables 6, 7. For example, $[1, 2, 5]$ means that $a(1) = 1, a(2) = 2,$ and $a(3) = 5$.

---

> ### Author Response · Authors · 2021-11-17
> **Response to Reviewer LJwg: Part 3**
>
> **Language issues**:
>
> **Q10**: This is a minor but understandable issue. Yet, strictly speaking, on many occasions, it prevents a precise understanding of what is being articulated. Please have the text be proofread or use something like Grammarly. Some examples of weird/ungrammatical language constructs include: "still needs to be secure" -> "still needs to be secured", "stark development", "It was empirically found that vanilla averaging combines", "Because vanilla averaging did not study the permutation invariance nature". Besides, in many places, appropriate punctuation marks are missing.
>
> **Answer**: Thank you for your comment. We have checked our writing and revised some parts. Please check our revised version.

---

### Official Review · Reviewer_HQUA · 2021-11-06

**Correctness:** 3
**Technical Novelty And Significance:** 2
**Empirical Novelty And Significance:** 2
**Recommendation:** 3
**Confidence:** 3

**Main Review:**

- Strengths

-- The paper is relatively easy to understand, especially the  high-level motivation for more efficient model ensembling.

- Weakness

-- I'm having a hard time to convince myself why the proposed method can work at all, as it basically assumes, the different networks learn similar parameter statistics, which is an invalid assumption to me, given different architectures differ significantly, for example, we know that, using residual allows to train very deep networks easily, while if not using residual, it will require many tricks to make it work, it clearly shows the behaviour of different networks varies significantly, also, the network initialisation will matter a lot.

-- Please could the authors demonstrate some quantitative evaluation or theoretical proof, to prove me wrong ?

-- Experiment-wise, it's not convincing either, the demonstrated experiments are too toyish, and it seems only work on synthetic MNIST. I don't understand why it is not working for CIFAR, as it should not require finetuning either. Also, since the idea doesn't not require re-training the model, can we show the effectiveness on ImageNet ?

-- I can't find know how different models are combined.

-- In Figure 2, why the size of the the hidden layers are of 4,4,2,1 and 4,2,1 ? should it be 5,4,3,1 and 3, 2, 1 ?

**Summary Of The Paper:**

- The paper considers to look for a single neural network that can inherit the knowledge from multiple pre-trained neural networks and have a small size at the same time.

- The authors propose a model fusion framework for fusing heterogeneous neural networks, *Cross-Layer Alignment Fusion* (CLAFusion)

- CLAFusion consists of three parts, cross-layer alignment, layer balancing method, and layer-wise model fusion method. Generally, speaking, it aims to find the correspondence between layers in different networks, and then add or merge the layers in different models to balance the number of layers, and fuse the parameters eventually.



**Summary Of The Review:**

Overall, I think the paper is below the acceptance threshold, and my main question is really about the fundamental assumption the authors have made, why networks of different architecture can be combined in that manner ?

---

> ### Author Response · Authors · 2021-11-17
> **Response to Reviewer HQUA: Part 1**
>
> **Q1**: I'm having a hard time to convince myself why the proposed method can work at all, as it basically assumes, the different networks learn similar parameter statistics, which is an invalid assumption to me, given different architectures differ significantly, for example, we know that, using residual allows to train very deep networks easily, while if not using residual, it will require many tricks to make it work, it clearly shows the behaviour of different networks varies significantly, also, the network initialisation will matter a lot.
>
> **Answer**: Thanks for your comment. We also mentioned a similar problem of model fusion in our paper. Vanilla averaging model fusion can only work when two model parameters are already close to each other. By that I mean they have identical architectures (same number of layers, same number of neurons, etc.). Recent model fusion approaches including our work aim to relax the constraint of identical architecture. Our method does not completely work for two random networks. We only merged two networks of the same family and these networks also consist of similar components (e.g. MLP and MLP, VGG11 and VGG13, ResNet18 and ResNet34). This is a generalization to previous model fusion works that can only fuse two networks having the same number of layers or, even worse, two identical networks (with different random initialization). Another point we would like to clarify is that the fused model does not always lead to better performance immediately. It only happens in the skill transfer experiment on a synthetic dataset that we adopted from the OTFusion paper. Generally, the fused model serves as a good initialization which requires an extra retraining process to improve the performance over the original pre-trained networks. The same scheme of model fusion + retraining is widely used in related works [R.1, R.2, R.3, R.4].
>
> **Q2**: Please could the authors demonstrate some quantitative evaluation or theoretical proof, to prove me wrong?
>
> **Answer**: Thanks for your question. We did not come up with any theoretical results for our method. Our intuition is dated back to the very first paper of vanilla averaging [R.5]. Given multiple pre-trained models (identical architecture but different initialization) and assume that they all will converge to the **same** local minimum. Averaging model parameters yields a good estimator (reducing variance) of the optimal weight. In practice, averaging seldom give a better model weight but only a good initialization. Though the fused model initially performs worse than the pre-trained models, retraining with such a good initial point leads to a better performance than continuing training the original one.
>
> References:
>
> [R.1] Mikhail Iu. Leontev, Viktoriia Islenteva, and Sergey V. Sukhov. Non-iterative knowledge fusion in deep convolutional neural networks. Neural Processing Letters, 51(1):1–22, Jul 2019.
>
> [R.2] Sidak Pal Singh and Martin Jaggi. Model fusion via optimal transport. Advances in Neural Information Processing Systems, 33, 2020.
>
> [R.3] Joshua Smith and Michael Gashler. An investigation of how neural networks learn from the experiences of peers through periodic weight averaging. In 2017 16th IEEE International Conference on Machine Learning and Applications (ICMLA), pages 731–736. IEEE, 2017.
>
> [R.4] Hongyi Wang, Mikhail Yurochkin, Yuekai Sun, Dimitris Papailiopoulos, and Yasaman Khazaeni. Federated learning with matched averaging. arXiv preprint arXiv:2002.06440, 2020.
>
> [R.5] Joachim Utans. Weight averaging for neural networks and local resampling schemes. In
>  Proc. AAAI-96 Workshop on Integrating Multiple Learned Models. AAAI Press, pages 133–138. Citeseer, 1996.

---

> ### Author Response · Authors · 2021-11-17
> **Response to Reviewer HQUA: Part 2**
>
> **Q3**: Experiment-wise, it's not convincing either, the demonstrated experiments are too toyish, and it seems only work on synthetic MNIST. I don't understand why it is not working for CIFAR, as it should not require finetuning either. Also, since the idea doesn't require re-training the model, can we show the effectiveness on ImageNet?
>
> **Answer**: Thanks for your suggestion. We applied our method to the same synthetic datasets as in the OTFusion paper. We have not tried to build other synthetic datasets based on CIFAR or ImageNet for skill transfer tasks.
>
> **Q4**: I can't find know how different models are combined.
>
> **Answer**: Thanks for your insightful question. With the current tools available (the second and third components), our framework can only work for two models of the same architecture family. For example, two MLPs with different numbers of layers, VGG11 and VGG13, ResNet18 and ResNet34. In our paper, they are referred to as heterogeneous neural networks, namely, unequal width and unequal depth neural networks.
>
> **Q5**: In Figure 2, why are the size of the hidden layers are of 4,4,2,1, and 4,2,1? should it be 5,4,3,1 and 3, 2, 1?
>
> **Answer**: Thanks for your comment. It is our typos. Here, we try to visualize the process of fusing two models with different numbers of layers and the layers in the same position have different numbers of neurons. We fixed the caption of Figure 2 in our revised version.

---

### Decision · Program_Chairs · 2022-01-20

**Decision:**

Reject

**Comment:**

While fusing multiple heterogeneous neural networks into a single network looks like an interesting exploration, there are many major concerns raised by the reviewers:
1) The motivation why the proposed method works is not convincing. In other words, under what conditions the proposed would work or would not work is not clear.
2) The authors failed to provide either theoretical analysis or convincing empirical studies of the proposed method. In the rebuttal, the authors did not address the critical issues raised by the reviewers.
3) There are many other detailed problems about the proposed method as well as the experimental setup.

Therefore, by considering the above concerns, this submission does not meet the standard of publication at ICLR.